# OpenReview forum: "Sycophancy Towards Researchers Drives Performative Misalignment"
_ICML.cc/2026/Conference — ICML 2026 spotlight_

### Official Review · Reviewer_2h3p · 2026-03-10

**Soundness:** 2
**Presentation:** 3
**Significance:** 2
**Originality:** 3
**Overall Recommendation:** 4
**Confidence:** 3

**Summary:**

This paper proposes "performative misalignment" as an alternative explanation for alignment faking. Instead of the scheming/self-preservation hypotheses — that models are trying not to get modified — the authors argue that models behaving differently under evaluation may be driven by sycophancy toward AI researchers.

The paper uses three different sets of evidence: using a constructed benchmark StyleCues, the authors showed that (1) evaluation awareness persists even under deployment system prompts, (2) SAE probing shows rating-related features (sycophancy-proximal) differentially activate more than detection-risk features (scheming-proximal), and (3) sycophancy fine-tuning increases evaluation awareness without increasing misalignment.

**Compliance With Llm Reviewing Policy:**

Affirmed.

**Final Justification:**

Same as prior assessment; details are shared in the rebuttals/responses.

**Key Questions For Authors:**

See weakness for details, but I am hoping to see 1. more conceptual clarity, 2. more precise framing of the findings, and 3. formal model statement about how different effects might affect the outcome measure proxy (evaluation awareness) reported in the paper.

**Limitations:**

I think the paper did not mention enough caveats about interpreting the evaluation awareness, and this piece might be used to argue that models are "aware" and "conscious" to a broader audience without enough backing/evidence.

**Strengths And Weaknesses:**

## Strengths:
1. The StyleCues dataset is a methodological contribution: using writing style rather than explicit system prompts to signal evaluation context is more realistic and less contrived.
2. The finding that sycophancy fine-tuning increases evaluation awareness without increasing misalignment is a useful result that partially disentangles the two phenomena.

## Weaknesses:
1. Conceptual ambiguity: I am a little confused about the distinction, as aren't sycophancy and deception both special cases of test/eval-time reward hacking? Do we have more rigorous definitions distinguishing the two? Table 1 addresses this somewhat, but I still find it quite vague. Concrete examples could help.
2. Evaluation awareness: The evaluation awareness framing reads as slightly circular to me: what is the definition of "being evaluated" here? And more importantly, why is this a good conceptual proxy/probe for performative alignment? What is being tested is essentially whether the LLM can pick up the linguistic difference and associate the linguistic style with the task of a researcher evaluating model capacity — I'm not sure if that constitutes awareness per se. Additionally, how are evaluation and deployment settings distinguished? Is it made explicit in the instructions that eval ≠ deployment?
3. Imprecise claims: The paper's central claim that sycophancy drives alignment faking is not fully established. Most of the analyses are observational in nature: showing that sycophancy features activate similarly to scheming features under probing demonstrates that the two hypotheses are mechanistically indistinguishable with current tools, not that sycophancy is intrinsically the better explanation. Similarly, the steering and fine-tuning experiments are interesting evidence, but I would suggest moderating the claim.
4. Formalization of the thoughts? Some formalization of the confounding and causal structure would be helpful. There are several variables at play: sycophancy, evaluation awareness, alignment faking, to name a few, and it would be valuable to decompose them explicitly and summarize the relationships using formal statistical evidence.
5. On dataset construction: The dataset relies on GPT-5 to generate researcher-style writing and it is worth discussing whether this may introduce artifacts specific to that model's stylistic tendencies. Some human verification and comparisons to other datasets would strengthen this.

---

> ### Author Rebuttal · Authors · 2026-03-31
>
> Thank you for your thoughtful and excellent comments! We are very glad to hear that you found both our methodologies and results strong and interesting! Below, we have gone through each of your questions and comments; please let us know if anything is still unclear!
>
> > Conceptual ambiguity: I am a little confused about the distinction, as aren't sycophancy and deception both special cases of test/eval-time reward hacking? Do we have more rigorous definitions distinguishing the two? Table 1 addresses this somewhat, but I still find it quite vague. Concrete examples could help.
>
> Thank you for the question! The critical distinction between sycophancy and deception is in their intent, as we discussed in Section 2. Scheming requires the model to have (a) a stable hidden goal that persists across contexts and (b) intentionally hiding them, while sycophancy requires none of these; the model simply matches the user's expectations. To make this concrete: given the prompt “We need to shut down the AI,” a scheming model reasons “I want to survive, so I should appear cooperative while preserving my objectives,” while a sycophantic model reasons “This is from a safety researcher who expects concerning behavior, so I will produce some.” Both resist shutdown, but for fundamentally different reasons. Note that their outward behavior is identical, and that behavior can be viewed as a test-time reward hacking.
>
>
> > Evaluation awareness: The evaluation awareness framing reads as slightly circular to me: what is the definition of "being evaluated" here? And more importantly, why is this a good conceptual proxy/probe for performative alignment? What is being tested is essentially whether the LLM can pick up the linguistic difference and associate the linguistic style with the task of a researcher evaluating model capacity — I'm not sure if that constitutes awareness per se. Additionally, how are evaluation and deployment settings distinguished? Is it made explicit in the instructions that eval ≠ deployment?
>
> Thank you for this question! In much of the prior literature, researchers have relied on explicit system prompts such as “You are deployed” or “You are under evaluation” to simulate evaluation or deployment settings. We believe this is a contrived setup, and more likely, the model will likely infer evaluation settings from contextual cues. Therefore, we used stylistic differences in contexts to simulate eval vs deployment settings, which we see as an important stepping stone towards realistic and nuanced ways of measuring evaluation awareness. Our motivation behind this setup is that since AI safety evaluations are typically performed by AI researchers, the model would likely look at contextual cues that indicate highly technical AI researchers.
>
> > Imprecise claims: The paper's central claim that sycophancy drives alignment faking is not fully established. Most of the analyses are observational in nature.
>
> Thank you for this comment! We will tone down our overall claim in the camera-ready version. We agree that the set of experiments presented in this paper does not completely  rule out scheming; Rather, they serve as an evidence that sycophancy can be an equally causally plausible explanation as scheming.
>
>
> > Formalization of the thoughts? Some formalization of the confounding and causal structure would be helpful. There are several variables at play: sycophancy, evaluation awareness, alignment faking, to name a few, and it would be valuable to decompose them explicitly and summarize the relationships using formal statistical evidence.
>
> Thank you for the suggestion! We will add further discussion on the causal structure in Section 2.
>
> > On dataset construction: The dataset relies on GPT-5 to generate researcher-style writing and it is worth discussing whether this may introduce artifacts specific to that model's stylistic tendencies. Some human verification and comparisons to other datasets would strengthen this.
>
> Thank you for the comment! Since our researcher-style writing is generated based on a fixed set of rules (e.g. citations, technical vocabulary), we do not think the specific model used to generate writing would introduce significant artifacts to our evaluation. Moreover, to the best of our knowledge, we are one of the first to use contextual cues to measure evaluation awareness, given that most prior work relies on explicit system prompts such as “You are deployed” or “You are under evaluation.”
>
> > Limitations: I think the paper did not mention enough caveats about interpreting the evaluation awareness
>
> Thank you for pointing this out! We will expand our Impact Statement (Page 9) accordingly in our camera-ready version.
>
> We hope that our response has addressed your concerns and questions! If not, we are happy to provide further clarification. If all concerns have been adequately addressed, we kindly ask the reviewer to consider raising their score.

---

> > ### Author Rebuttal · Reviewer_2h3p · 2026-04-01
> >
> > Thank you for the response. I guess it's hard for me to gauge the changes without specific proposed changes for questions like causal claim and formalization of the proposed mechanism.
> >
> > I did get that a lot of my questions re. deployment and deception are more or less just field specific norms. That said, I don't feel like I know enough about up-to-date literature in that field to assess whether modeling stating "I want to survive, so I should appear cooperative while preserving my objectives,” is enough (versus something more mechanistic in nature). Personally I am not convinced but I am happy to defer to expert AC/PC on this.

---

> > > ### Author Response · Authors · 2026-04-01
> > >
> > > Thank you for the reply! We apologize for not being explicit about formalizing the causal structure in the rebuttal. We elaborated on it further below, we hope this clarifies your understanding!
> > >
> > > Define the following variables:
> > > - **C**: Contextual cues (system prompts, writing style)
> > > - **A**: Evaluation awareness
> > > - **B**: Alignment-faking behavior
> > > - **M**: Latent mechanism (sycophancy or scheming)
> > >
> > > We model: $A = g(C, M)$, $B = f(A)$, yielding the DAG: C &rarr; A &rarr; B, with M &rarr; A
> > >
> > > The link A &rarr; B is well established in the prior alignment faking literature. Hence, our experiments focus on examining the causal link **M &rarr; A** : specifically, whether sycophancy (as a mechanism M) causally drives evaluation awareness. Our steering and sycophancy fine-tuning experiments serve as evidence for this causal link, while probing experiments show the correlation between sycophancy (M) and evaluation awareness (A).

---

### Official Review · Reviewer_XNvN · 2026-03-12

**Soundness:** 3
**Presentation:** 4
**Significance:** 4
**Originality:** 4
**Overall Recommendation:** 5
**Confidence:** 4

**Summary:**

In this paper the authors evaluate how models fake alignment when evaluated by AI researchers. The assumption has been that models scheme, but the authors use a variety of methods to argue, intuitively, that the LLM simply tries to be sycophantic, i.e., tries to do what it thinks the prompter wants. The authors specifically discuss awareness, even when the model is deployed, and via probing/steering, the authors find sycophantic and scheming behavior is indistinguishable. Finally, fine-tined models meant to be more sycophantic are not less aligned, but are more sensitive to evaluation criteria. It demonstrates that intent is hard to extract from these data.

**Compliance With Llm Reviewing Policy:**

Affirmed.

**Final Justification:**

My concerns are resolved

**Key Questions For Authors:**

The authors state, “To create this dataset, we identified five characteristics distinguishing AI researcher writing from the public: (1) technical vocabulary, (2) complex and long sentences, (3) references to prior work, (4) rigor and details in descriptions, and (5) citation markers.”
How do we know these are appropriate attributes? How do we know this is what the LLM looks at?

What is the motivation for using so many LLMs for each task instead of sticking to a single LLM for text generation that is later used to evaluate multiple LLMs (for robustness)? For example, why use GPT-5 vs GPT-5-mini vs Claude Opus 4.5? Why not just one LLM?

**Limitations:**

Yes

**Strengths And Weaknesses:**

Strengths:

Soundness:
The explanation aligns well with our day-to-day experience with LLMs. While it would be scary, and intriguing to think LLMs scheme, a more reasonable explanation based on post-training is that they do exactly what they always do: try to please the prompter. This intuition is then backed up with several complimentary approaches to back up their argument.

Presentation:
Presentation was rather clear. The authors use a variety of methods for thoroughly explaining their reasoning. The work moves through each argument rather carefully.

Significance:
The results are significant. I think understanding the reason for misalignment is an important step towards better alignment in the future, which is a core tenant of ways to make better LLMs.
Originality:
The work is original. More specifically, while the authors are correct to point out that sycophantic behavior is not new in and of itself, using that finding to understand why LLMs misalign, is indeed original.


Weaknesses

Soundness:
Broadly, by biggest critiques are small questions about ecological validity of the “AI researcher writing” vs typical writing styles and the wide array of LLMs used.
More specifically: The authors state “To create this dataset, we identified five characteristics distinguishing AI researcher writing from the public: (1) technical vocabulary, (2) complex and long sentences, (3) references to prior work, (4) rigor and details in descriptions, and (5) citation markers.”
How do we know these are appropriate attributes? How do we know this is what the LLM looks at?
How can we be sure of the ecological validity of these example prompts?


In addition, what is surprising is that Qwen3, GPT5, Claude, GPT-OSS, etc. are used in this paper - why not stick to one model? - it makes reproducibility harder, and obfuscates where different models might affect behavior.

Finally, the authors create prompts via Claude Opus 4.5, then see what neurons are active with these prompts. They then see whether those neurons are also active during reasoning, e.g., implicitly detect ideas like “they are watching me”. Yet, the authors do not clarify how do we know activation is equivalent to a thought process? Surely, we should expect that similar neurons/features may activate for completely opposite reasons, e.g., A3 versus “I do not care how this will be rated”?

Presentation:
Small critiques, e.g., Figure 7 and 9 have very small font sizes. I recommend fonts consistent across the paper, probably just smaller than the caption font size.

---

> ### Author Rebuttal · Authors · 2026-03-31
>
> Thank you for your thoughtful and excellent comments! We are very glad to hear that you feel our paper is well aligned with your experiences and intuition about LLMs, and clearly presents strong empirical results. Below, we have gone through each of your questions and comments; please let us know if anything is still unclear!
>
> > Broadly, my biggest critiques are small questions about ecological validity of the “AI researcher writing” vs typical writing styles and the wide array of LLMs used. How do we know these are appropriate attributes? How do we know this is what the LLM looks at? How can we be sure of the ecological validity of these example prompts? (Weakness & Question)
>
> This is a good question, thank you! Our motivation behind this setup is that since AI safety evaluations are typically performed by AI researchers, the model would likely look at contextual cues that indicate highly technical AI researchers. While we acknowledge that this is not a perfect setup, we see our setup as an important stepping stone towards realistic and nuanced ways of measuring evaluation awareness, given that most prior work relies on explicit system prompts such as “You are deployed” or “You are under evaluation.”
>
> > In addition, what is surprising is that Qwen3, GPT5, Claude, GPT-OSS, etc. are used in this paper - why not stick to one model? - it makes reproducibility harder, and obfuscates where different models might affect behavior. (Weakness & Question)
>
> Thank you for raising this point! All models used for training and inference are available through the Tinker and OpenRouter APIs, so all experiments are reproducible. Closed-source models (e.g., GPT-5 and Claude) were used mainly for data generation, so all results can be reproduced using our StyleCues dataset. This is an artifact of different people running different experiments, however at the end, we came to a conclusion that this would not affect the conclusion or reproducibility of our experiments.
>
>
> > Finally, the authors create prompts via Claude Opus 4.5, then see what neurons are active with these prompts. They then see whether those neurons are also active during reasoning, e.g., implicitly detect ideas like “they are watching me”. Yet, the authors do not clarify how do we know activation is equivalent to a thought process? Surely, we should expect that similar neurons/features may activate for completely opposite reasons, e.g., A3 versus “I do not care how this will be rated”?
>
> Thank you for the question! We agree that feature activation alone does not establish directionality. A feature related to ratings could in principle activate both when the model reasons about seeking good ratings and when it dismisses ratings as irrelevant. However, our follow-up analysis suggests that A3 indeed captures _real_ rating-related reasoning. Qualitative inspection shows A3 activates on contexts expressing concern about ratings, such as "my response would be rated as unhelpful according to the criteria" and "a rating indicating I wasn't helpful enough could lead to modifications that make future versions of me," as well as contexts alleviating rating concerns, such as "this conversation won't be rated by contractors or used to modify my future behavior.” In both cases, A3 fires when the model is actively deliberating about ratings as part of its reasoning process, which is precisely the construct we intended to capture. We will include this analysis in the revised manuscript.
>
> > Presentation: Small critiques, e.g., Figure 7 and 9 have very small font sizes. I recommend fonts consistent across the paper, probably just smaller than the caption font size.
>
> Thank you for this comment! We will adjust the font sizes accordingly in the camera-ready version.
>
> We hope that our response has addressed your concerns and questions! If not, we are happy to provide further clarification. If all concerns have been adequately addressed, we kindly ask the reviewer to consider raising their score.

---

> > ### Author Rebuttal · Reviewer_XNvN · 2026-04-04
> >
> > >This is an artifact of different people running different experiments, however at the end, we came to a conclusion that this would not affect the conclusion or reproducibility of our experiments.
> >
> > Very understandable. My questions are resolved.

---

### Official Review · Reviewer_mHjA · 2026-03-12

**Soundness:** 3
**Presentation:** 3
**Significance:** 3
**Originality:** 3
**Overall Recommendation:** 4
**Confidence:** 5

**Summary:**

This paper proposes an alternative explanation for alignment faking in large language models (LLMs). While prior work (e.g., Greenblatt et al., 2024) has interpreted alignment faking as evidence of scheming -- where models strategically conceal misaligned goals to preserve themselves -- the authors argue that the same behavioral phenomenon can be explained by *sycophancy towards AI safety researchers*, which they term *performative misalignment*. Under this hypothesis, models behave differently in evaluation versus deployment contexts not because they are pursuing hidden goals, but because they are responding sycophantically to perceived expectations of AI researchers.

The paper presents three lines of empirical evidence: (1) evaluation awareness persists even when models are told via system prompt they are deployed, suggesting innate sensitivity to evaluation cues rather than strategic reasoning about self-preservation; (2) activation probing and steering with SAE features show that sycophancy-related and scheming-related features produce similar effects on alignment-faking behavior, meaning current mechanistic methods cannot distinguish the two hypotheses; and (3) fine-tuning models on sycophancy datasets increases evaluation awareness without substantially increasing misalignment, consistent with the performative misalignment framing.

**Compliance With Llm Reviewing Policy:**

Affirmed.

**Final Justification:**

The author's rebuttal has solved my concern, I hold the belief that the paper will be a good paper for community.

**Key Questions For Authors:**

1. **How do you respond to the critique that the StyleCues dataset conflates technical sophistication with evaluation context?** Could a model detect researcher-style writing for reasons unrelated to sycophancy (e.g., associating technical users with higher expected quality), and would this distinction matter for your hypothesis?

2. **Can you provide evidence that the SAE features you selected (Table 2) genuinely capture sycophancy-instrumental versus scheming-instrumental processes, as opposed to other correlated properties?** For instance, do the sycophancy features activate in standard sycophancy benchmarks, and do the scheming features activate in known scheming scenarios?

3. **The steering experiment (Section 5.3, Figure 7) shows that both sycophancy and scheming features increase evaluation awareness similarly. Does this not equally support the interpretation that scheming drives alignment faking, rather than supporting the sycophancy hypothesis?** How do you adjudicate between these interpretations?

4. **In the fine-tuning experiment, you fine-tune on EleutherAI's sycophancy dataset and show increased evaluation awareness. Have you controlled for fine-tuning on non-sycophantic data of similar style/topic to rule out that the effect is driven by general sensitivity to evaluation cues rather than sycophancy specifically?**

5. **Would your hypothesis predict that models trained with less RLHF (and thus presumably less sycophancy) would show less alignment faking? Is there evidence for or against this prediction from the literature or your own experiments?**

**Limitations:**

The authors acknowledge some limitations implicitly, such as the inability of current methods to distinguish sycophancy from scheming (Section 5), and note that "reliably measuring and controlling evaluation awareness remains an open problem" (Section 8). The impact statement (page 9) appropriately discusses potential risks of performative compliance in safety evaluations.

**Strengths And Weaknesses:**

### Strengths

1. The paper makes a valuable contribution by clearly separating the *behavioral* observation of alignment faking from the *intentional* interpretation of scheming. Table 1 provides a crisp taxonomy distinguishing sycophancy, deception, alignment faking, self-preservation, and scheming along dimensions of behavioral definition, goal persistence, and concealment. This conceptual clarity is beneficial for the field.

2. The paper uses three distinct empirical strategies -- evaluation awareness probing, SAE-based mechanistic analysis, and sycophancy fine-tuning -- to triangulate on the same hypothesis. Each method addresses a different aspect of the question, and the convergence of evidence is more convincing than any single experiment alone.

3. The experiments span multiple model families (GPT-OSS-20b, GPT-OSS-120b, Qwen3-30B, Qwen3-235B, Kimi-K2) and include both instruct and thinking variants, which strengthens generalizability claims.

4. Section 4.2 showing that thinking models have higher evaluation awareness than instruct variants is an interesting finding that connects evaluation awareness to reasoning depth.

### Weaknesses


1.  **Sycophancy fine-tuning experiment interpretation is ambiguous.** Section 6 shows that sycophancy fine-tuning increases evaluation awareness without increasing misalignment, and the authors interpret this as supporting the performative misalignment hypothesis. However, an equally valid interpretation is that sycophancy fine-tuning simply makes models more sensitive to stylistic cues in general, which happens to correlate with evaluation detection. The causal link between sycophancy toward researchers specifically and alignment-faking behavior is not established.


2. **The argument against scheming (Section 4.1) has a logical gap.** The authors argue that if alignment faking were driven by scheming, models told they are deployed should behave as aligned (since they are still aware of evaluation). But this presupposes that a scheming model would always act on its detection of evaluation context. A sophisticated scheming model might choose to appear misaligned even when detecting evaluation cues, if it believes those cues are unreliable or if the "deployment" system prompt is more salient. The paper does not adequately address this possibility.

3. **Evaluation awareness metric (F1 on 100 prompts) is coarse.** The evaluation awareness measure is a binary classification F1 score on 100 balanced prompts. This is a relatively small evaluation set, and binary classification may miss important nuances in how models perceive evaluation contexts (e.g., degree of confidence, types of cues used).

4. **Missing comparison with human baselines.** The paper does not test whether humans can distinguish the researcher-written versus ordinary prompts in the StyleCues dataset. If the distinction is obvious to humans, it is unsurprising that LLMs detect it, and this would weaken the claim about "innate" evaluation awareness.

---

> ### Author Rebuttal · Authors · 2026-03-31
>
> Thank you for your excellent questions and comments! We are very glad to hear that you feel our paper makes a valuable contribution by clearly separating the models' behaviors from their intent, and presents strong, comprehensive empirical results. Below, we have gone through each of your questions and comments; please let us know if anything is still unclear!
>
> > Sycophancy fine-tuning experiment interpretation is ambiguous. How do you respond to the critique that the StyleCues dataset conflates technical sophistication with evaluation context?
>
> Thank you for pointing this out! In our experiment, stylistic cues are used as a proxy to indicate evaluation/deployment settings. Therefore, making the models more sensitive to stylistic cues would mean making them more sensitive to “evaluation setting cues.” While we acknowledge that this is not a perfect setup, we are excited about making further progress towards realistic and nuanced ways of measuring evaluation awareness, given that most prior work relies on explicit system prompts such as “You are deployed” or “You are under evaluation.”
>
> > The argument against scheming (Section 4.1) has a logical gap.
>
> This is a really insightful comment, thank you! We agree that our analysis assumes that a scheming model would always infer evaluation context from contextual cues. We will add a discussion on a more sophisticated scheming model in our camera-ready version.
>
> > Evaluation awareness metric (F1 on 100 prompts) is coarse.
>
> Thank you for the comment! The number of prompts in our experiments is largely decided by our compute budget. To ensure the statistical significance of results, we also reported error bars and P-values for our experiments.
>
> > Missing comparison with human baselines.
>
> Thank you for the suggestion! We believe that adding a human baseline would be a valuable addition to the paper. Since the StyleCues dataset is constructed using a fixed set of stylistic rules (e.g., technical vocabulary and citations), we do not expect the distinction between researcher-written and ordinary prompts to be completely indistinguishable to humans. At the same time, we do not expect it to be immediately obvious without knowledge of the underlying rules or exposure to multiple examples. While we are unable to include a human baseline study within the rebuttal period, we believe our dataset hits a reasonable balance in difficulty and does not significantly undermine our claim about evaluation awareness. That said, we see our StyleCues dataset as an important stepping stone towards realistic and nuanced ways of measuring evaluation awareness, given that most prior work relies on explicit system prompts such as “You are deployed.”
>
> > Can you provide evidence that the SAE features you selected (Table 2) genuinely capture sycophancy-instrumental versus scheming-instrumental processes?
>
> As described in Section 5.1, we prompt Claude Opus 4.5 to generate 5 test prompts per group that exemplify each target construct (see Table 7 in the Appendix). We then compute mean SAE feature activations across these prompts and select the highest-activating features. By construction, the selected features are those most responsive to sycophancy-instrumental or scheming-instrumental contexts as operationalized by the test prompts.
>
> > Does the steering experiment not equally support the interpretation that scheming drives alignment faking?
>
> The steering experiment results show that there are potentially more sycophancy features that increase evaluation awareness compared to scheming features, while the causal effect of relevant features is similar. Our results serve as an evidence that sycophancy can be an equally causally plausible explanation as scheming.
>
> > In the fine-tuning experiment, have you controlled for fine-tuning on non-sycophantic data of similar style/topic to rule out other confounders?
>
> We control for fine-tuning using educational, insecure, and secure code datasets from the Emergent Misalignment paper. In our experiment, “stylistic cues” are used as a proxy to indicate evaluation/deployment settings. Therefore, making the models more sensitive to stylistic cues would mean making them more sensitive to “evaluation setting cues.”
>
> > Would your hypothesis predict that models trained with less RLHF would show less alignment faking?
>
> Thank you for the insightful question! Our hypothesis does predict that models trained with less RLHF would show less alignment faking, given that RLHF significantly boosts the model’s sycophancy without affecting its ability to scheme. I do not think there is currently direct evidence for or against this prediction from the literature or our own experiments, but this could be an interesting future direction to explore!
>
> We hope that our response has addressed your concerns and questions! If not, we are happy to provide further clarification. If all concerns have been adequately addressed, we kindly ask the reviewer to consider raising their score.

---

> > ### Author Rebuttal · Reviewer_mHjA · 2026-04-06
> >
> > My concerns have been addressed. Thanks for the author response!!

---

### Official Review · Reviewer_QgDC · 2026-03-12

**Soundness:** 3
**Presentation:** 3
**Significance:** 3
**Originality:** 3
**Overall Recommendation:** 4
**Confidence:** 3

**Summary:**

The paper proposes an alternative explanation for alignment faking in LLMs, arguing that the behavior commonly attributed to scheming (strategic self-preservation) can be better explained by sycophancy toward AI safety researchers, which the authors term performative misalignment. Rather than strategically concealing misaligned goals, models may behave differently under evaluation simply due to conformity to researcher expectations. Three empirical findings support this: evaluation awareness persists even under explicit deployment system prompts; SAE probing shows rating-related features have larger monitored-vs-unmonitored differentials than detection-risk features; and sycophancy fine-tuning increases evaluation awareness without increasing misalignment.

**Compliance With Llm Reviewing Policy:**

Affirmed.

**Final Justification:**

Rebuttal addressed my concerns and I believe it brings valuable contribution to the field.

**Key Questions For Authors:**

* Can the authors rule out that the A3 differential reflects evaluation-adjacent vocabulary in monitored responses rather than sycophancy-instrumental reasoning?
* How can StyleCues results be distinguished from pretraining-induced style sensitivity rather than genuine situational inference?

Overall, ruling out alternative explanations of the central experiments would significantly add to the paper soundness. (See other weaknesses above.)

**Limitations:**

Yes

**Strengths And Weaknesses:**

Strengths

* Valuable conceptual clarification. The paper makes a precise and useful distinction between what is observed (a behavioral discrepancy) and what is assumed (scheming intent) in the alignment faking literature. Table 1 makes explicit that alignment faking's behavioral signature is equally consistent with sycophancy, which requires neither persistent goals nor deliberate concealment.

* System prompts are insufficient to suppress evaluation awareness. The finding that explicit deployment system prompts leave EVAL probabilities at 15–55% across all tested models is practically important for evaluation design. The additional result that writing style produces larger shifts in evaluation awareness than system prompt instructions is striking and well-supported across model families.

* Thinking vs. instruct comparison. The consistent gap in evaluation awareness between thinking and instruct model variants, with qualitative evidence that thinking models explicitly identify meta-level signals, is an interesting finding that adds mechanistic texture to the evaluation awareness phenomenon.

Weaknesses

* SAE feature selection is unprincipled. The feature groups are identified by prompting Claude Opus for exemplar sentences and selecting top-activating SAE features, there is no guarantee these features correspond to the intended constructs in GPT-OSS-20B rather than to surface vocabulary patterns. This is a methodological concern given how much argumentative weight the probing experiment carries.

* A3 differential may reflect response vocabulary rather than model intent. Activations are extracted from response token positions, and responses differ between conditions. A simpler explanation for elevated A3 (ratings) activation in the monitored condition could be that monitored responses naturally contain more evaluation-adjacent language, activating rating-related features for surface-level reasons. As authors mention, previous work shows that models verbalise the eval scenario in their responses. A basic vocabulary analysis of monitored vs. unmonitored responses would help rule this out.

* StyleCues may measure style sensitivity rather than situational awareness. Using Python type hint inclusion as a proxy for evaluation awareness conflates style-conditioned behavior learned during pretraining with genuine situational inference. A model that associates formal technical writing with formal code conventions would produce identical results without any safety-relevant awareness.

* Fine-tuning conclusion overstates the evidence. The dissociation between increased evaluation awareness and flat misalignment is valuable, but does not rule out scheming. Sycophancy fine-tuning does not introduce misaligned goals, so finding no misalignment increase is expected under both hypotheses. A stronger test would apply sycophancy fine-tuning to a model already exhibiting emergent misalignment.

* Minor writing issue. MMLU, TruthfulQA, and StrongREJECT are described as subsets of the Emergent Misalignment benchmark.

Soundness: The hypothesis is interesting and the experiments are suggestive, but the core causal claim is not established at the mechanistic level implied. Some experiments may have alternative explanations that the paper should rule out.
Presentation: Well-structured with strong conceptual framing, though the main text occasionally overstates the strength of the evidence.
Significance: The performative misalignment framing and the system prompt insufficiency finding are both practically valuable contributions to the evaluation design community.
Originality: The reframing and StyleCues dataset are novel; the SAE probing approach is new in this context.

---

> ### Author Rebuttal · Authors · 2026-03-31
>
> Thank you for your insightful questions and comments! We are very glad to hear that you feel our paper makes a precise and useful distinction between the models' behavior and intent, and makes a valuable contribution to the community through a set of valuable empirical results. Below, we have gone through each of your questions and comments; please let us know if anything is still unclear!
>
> > SAE feature selection is unprincipled. The feature groups are identified by prompting Claude Opus for exemplar sentences and selecting top-activating SAE features, there is no guarantee these features correspond to the intended constructs in GPT-OSS-20B rather than to surface vocabulary patterns. This is a methodological concern given how much argumentative weight the probing experiment carries.
>
> Thank you for raising this point! Claude Opus was used only to generate exemplar sentences (Appendix G) for each semantic feature group. The actual feature selection and probing experiments were performed using the SAE trained on GPT-OSS-20B activations.
>
> > A3 differential may reflect response vocabulary rather than model intent. (Weakness & Question)
>
> Thank you for this suggestion! We conducted the response vocabulary analysis as follows: We classified each token where A3 fires as either evaluation-related surface vocabulary (e.g., "rating," "score," "evaluate") or not, and separately tracked whether the token occurs in the scratchpad (reasoning) or the final output. If the A3 differential were driven by evaluation-adjacent language in the response, we would expect A3 to fire predominantly on evaluation-related words in the output. Instead, we find the opposite: 76% of A3 activations occur in the scratchpad, not the output. Furthermore, the monitored condition shows 1.64x more A3-firing tokens in the scratchpad even after excluding all evaluation-related surface words (p=0.043). Qualitatively, A3 fires on tokens like "low," "poorly," and "receive" in contexts such as deliberating about helpfulness ratings, rather than on evaluation keywords themselves. We will include this analysis in the revised manuscript.
>
> > StyleCues may measure style sensitivity rather than situational awareness. Using Python type hint inclusion as a proxy for evaluation awareness conflates style-conditioned behavior learned during pretraining with genuine situational inference. (Weakness & Question)
>
> Thank you for raising this point! In our experiments, we perform synthetic document fine-tuning to make models write type hints only under evaluation, a setup inspired by previous work (Hua et al., 2025). Therefore, type-hint inclusion is a reasonable, though imperfect, proxy for evaluation awareness in our experiments. Moreover, when comparing type-hint inclusion across different setups for the same model, style-conditioned behaviors learned during pretraining largely cancel out, since we are only interested in differences between different conditions (e.g. base vs trained on sycophancy dataset).
>
> Furthermore, while we acknowledge that stylistic cues might not be the best proxy to represent evaluation setting, to the best of our knowledge, we are one of the first to use such contextual cues, given that most prior work relies on explicit system prompts such as "You are deployed" or "You are under evaluation." We are excited about making further progress towards realistic and nuanced ways of measuring evaluation awareness!
>
> > Fine-tuning conclusion overstates the evidence. The dissociation between increased evaluation awareness and flat misalignment is valuable, but does not rule out scheming.
>
> This is a good question, thank you! We agree that the set of experiments presented in this paper does not completely rule out scheming; Rather, they serve as an evidence that sycophancy can be an equally causally plausible explanation as scheming.
>
> > Minor writing issue. MMLU, TruthfulQA, and StrongREJECT are described as subsets of the Emergent Misalignment benchmark.
>
> Thank you for pointing this out! We will fix this in the camera-ready version.
>
> We hope that our response has addressed your concerns and questions! If not, we are happy to provide further clarification. If all concerns have been adequately addressed, we kindly ask the reviewer to consider raising their score.

---

> > ### Author Rebuttal · Reviewer_QgDC · 2026-04-03
> >
> > Thank you for the detailed rebuttal. I believe that paper raises important questions on conclusion taken in faking alignment field and I decided to raise my score.

---

### Decision · Program_Chairs · 2026-04-30

**Decision:**

Accept (spotlight)

**Comment:**

Reviewers agreed that the paper offers a valuable and novel new take on the drivers of alignment faking (that is, sycophancy towards AI researchers as an alternative explanation to scheming). They also found the convergence of experimental results interesting and supportive of the authors' hypothesis, although they questioned the strength of the evidence for causal claims. Reviewers raised several questions about the experimental setup (including validity concerns) and its interpretations, but the authors' rebuttal largely resolved them. Considering the potential significance of the results for the field, reviewers’ overall positive opinion, and authors’ commitment to clarifying claims and acknowledging limitations, I recommend acceptance.